# Implications of a Soy-Based Diet for Animal Models

**DOI:** 10.3390/ijms22020774

**Published:** 2021-01-14

**Authors:** Justine Dhot, Valentine Prat, Marine Ferron, Virginie Aillerie, Angélique Erraud, Bertrand Rozec, Michel De Waard, Chantal Gauthier, Benjamin Lauzier

**Affiliations:** Université de Nantes, CHU Nantes, CNRS, INSERM, L’institut du Thorax, F-44000 Nantes, France; jc.dhot@gmail.com (J.D.); valentine.prat@gmail.com (V.P.); ferronmarine@gmail.com (M.F.); virginie.aillerie@univ-nantes.fr (V.A.); angelique.erraud@univ-nantes.fr (A.E.); bertrand.rozec@chu-nantes.fr (B.R.); Michel.Dewaard@univ-nantes.fr (M.D.W.); chantal.gauthier@univ-nantes.fr (C.G.)

**Keywords:** phytoestrogen, soy-based diet, sex, endothelium, diastolic function

## Abstract

The use of animal models in fundamental or pre-clinical research remains an absolute requirement for understanding human pathologies and developing new drugs. In order to transpose these results into clinical practice, many parameters must be taken into account to limit bias. Attention has recently been focused on the sex, age or even strain of each animal, but the impact of diet has been largely neglected. Soy, which is commonly used in the diet in varying quantities can affect their physiology. In order to assess whether the presence of soy can impact the obtained results, we studied the impact of a soy-based diet *versus* a soy-free diet, on diastolic function in a rat model based on transgenic overexpression of the β_3_-adrenergic receptors in the endothelium and characterized by the appearance of diastolic dysfunction with age. Our results show that the onset of diastolic dysfunction is only observed in transgenic male rats fed with a soy-free diet in the long term. Our study highlights the importance of the diet’s choice in the study design process, especially regarding the proportion of soy, to correctly interpret the outcome as low-cost diets are more likely to be highly concentrated in soy.

## 1. Introduction

Today, cardiovascular diseases are among the leading causes of death worldwide and account for 30% of global mortality, with 17.3 million deaths per year [1]. Understanding the pathophysiology of these diseases to identify new therapeutic targets is a major research objective. To achieve this, the use of animal models remains an important and necessary step in both basic and pre-clinical research and the choice of the animal model is a crucial step. In recent, years attention has been focused on the impact of sex [2], age [3] or strain of rodents [4] on the quality of the data obtained, whereas the impact of feeding has mostly been neglected. The majority of the used diets consist of soy, which is very popular because of its low cost, high protein contents, fibers and unsaturated fatty acids. It is also important to note that soy contains diadzein and genisteins, two isoflavones that are classified as endocrine disruptors [5]. Because of their structure they can act as agonists or antagonists of human estrogen receptors (ER), and are therefore referred to as phytoestrogens [6]. Because of their hormonal activity, isoflavones can interact with different signaling pathways, in particular the nitric oxide (^•^NO) pathway by modulating the transcription of different ^•^NO synthases (NOS) and can thereby indirectly modulate cardiovascular function. Given the potential strong effects on cardiovascular phenotypes, we decided to conduct a study to evaluate the impact of a long-term soy-based diet versus a soy-free diet in control rats (WT) and in a transgenic rat model (Tgβ_3_) characterized by the development of diastolic dysfunction in male.

## 2. Results and Discussion

Using echocardiography, we show that the systolic function based on the ejection fraction shows no change during aging in male, female or ovariectomized (OVX) female rats whatever the diet or the genotype (Figure 1A–C). The diastolic function was affected as shown by an increase in the E/A ratio (Figure 1D) for male Tgβ_3_ rats fed by soy-free diet only [7]. However, no changes in this parameter were observed in females, either OVX or not, fed with either soy-based or soy-free diets (Figure 1E,F). The diastolic alteration was confirmed by hemodynamic pressure analysis at 45 weeks of age in male Tgβ_3_ rats fed with the soy-free diet (Figure 1G). Echocardiography of female rats, with or without OVX, showed no alteration in their left ventricle (LV) and end-diastolic pressure (LVEDP) (Figure 1H,I). Our study, therefore, demonstrates that the presence of soy in diet can affect pathology development in a rodent model. Other studies have already investigated isoflavones, which is derived from soy, as a therapeutic tool in various pathologies (Xiao et al., 2018). On the other hand, Haines et al. reported that long-term consumption of phytoestrogens had no protective effect in females and was even fatal for males in a mouse model of hereditary hypertrophic cardiomyopathy. It is therefore important to take into account that soy levels in standard rodent diets induce phenotypic changes observed over a long period of time, but also through alteration in gene expression levels potentially via their action on ERs. In our study, the type of diet induced a change in the expression levels of NOS proteins. The expression of neuronal NOS (nNOS) was significantly lower in rats fed a soy-based diet (Figure 2A). While an increase in endothelial NOS (eNOS) expression was observed in rats fed with a soy-based diet and a significant decrease in rats fed with a soy-free diet (Figure 2B). The expression of inducible NOS (iNOS) is not altered by the genotype or the diet of the rats (Figure 2C). Acting as an estrogen receptor agonist or antagonist depending on the endogenous level of estrogen, isoflavones induce genomic or non-genomic effects that may be beneficial in pathological conditions [8]. Indeed, the administration of genistein reduces myocardial fibrosis and improves cardiac function and capillary density in the heart failure model [9]. Similar improvements were also observed in models of aortic constriction and volume overload. The mechanisms involved are thought to be related to the activation of the ^•^NO pathway by phosphorylation of eNOS or by activation of the transcription of the gene coding for eNOS. In addition to their antioxidant and cholesterol-lowering effect, isoflavones are also described as beneficial in breast cancer. However, isoflavones are also described as cancer stimulants. Thus, the beneficial or delayed effects of isoflavones are believed to be related to the duration of exposure to phytoestrogens. For example, one study showed that in utero and early exposure of rats to phytoestrogens resulted in increased resistance to volume overload [10,11]. Our results clearly indicate that diet choice is certainly not a trivial issue. Soy in the diet could therefore have a protective effect in case of diastolic dysfunction, one of the hypotheses being that the addition of isoflavones would play on the expression levels of eNOS and nNOS which would thus protect against the appearance of NOS-endothelial dysfunction described in male Tgβ_3_ rats. Beyond the question of the beneficial or negative effects of soy on cardiovascular function, these results highlight the importance of considering the potential impact of modifying dietary intake in animal models. Our study further raises an interesting avenue of reflection: it is possible that the development of diastolic dysfunction in Tgβ_3_ rats is dependent on the level of estrogens and phytoestrogens since only male animals, fed without phytoestrogens, develop this cardiac alteration. The fact that this cardiac dysfunction does not develop in rats fed with a soy-rich diet further indicates the role of hormones in this pathology. Whether we can extend this analogy to imagine a therapeutic approach to cases of heart failure with preserved ejection fraction, mainly characterized by diastolic dysfunction, remains to be investigated. 

## 3. Materials and Methods

### 3.1. Animal Model

We used a rodent model developed in our team: the transgenic (Tgβ_3_) rats overexpressing human β_3_-adrenergic receptor in endothelial cells, and their control (WT) rats. The transgenic rat model overexpressing hβ_3_-AR in endothelial cells was generated by microinjection of a linearized plasmid consisting of the hβ_3_-AR cDNA under the control of the endothelial cell-specific intercellular adhesion molecule-2 (ICAM-2) promoter into Sprague-Dawley zygotes [7]. Both male and female (OVX or not) rats were analyzed. The n values are shown on each histogram and vary from 3 to 22. All animals were fed, from 8 weeks of age, with a widely-used standard industrial diet containing soy (RM1, SDS diet) or with an equivalent soy-free diet (Envigo, #2914C, Huntingdon, UK) (Diet composition in Table 1). All animal experimental protocols were approved by the Pays de la Loire Ethical Committee and were performed in accordance with the French law on animal welfare, EU Directive 2010/63/EU for animal experiments, the National Institutes of Health (NIH) Guide for the Care and Use of Laboratory Animals (NIH Pub. No. 85-23, revised 2011), and the 1964 Declaration of Helsinki and its later amendments and all the animals were housed according to standard living conditions (protocol code 10060 and date of approval: 30/08/2017).

### 3.2. Echocardiography

Echocardiography was performed blindly at 15, 30 and 45 weeks of age on anesthetized rats using a Vingmed-General Electric (VIVID 7, Horten, Norway). The analyses were performed using off-line cineloop analysis software (Echopac TVI, GE-Vingmed Ultrasound) on the Therassay platform as previously described [12]. 

### 3.3. Pressure Measurement

At 45 weeks of age, rats were anesthetized with isoflurane and O_2_ under spontaneous ventilation. A pressure probe SPR 838 size 2F (Millar Instruments Inc., Houston, TX, USA) was inserted in the right carotid and pushed into the LVEDP was recorded using IOX1.5.7 software (EMKA Technologies, Paris, France).

### 3.4. Western Blots

After pressure measurement at 45 weeks of age, male rats were euthanized and hearts were excised. Total protein was extracted from LV powder as previously described [13]. Western blots were performed in order to evaluate the protein expression levels of nNOS (#4231S, Cell Signaling, Denvers, MA, USA), iNOS (AB5382, Millipore, Burlington, MA, USA), eNOS (610296, BD Biosciences, San Jose, CA, USA).

### 3.5. Statistical Analyses

Data are presented as mean ± SEM. For the comparisons involving two animal groups *of animals*, significances were defined using a Mann-Whitney test. For the echocardiography study, a one-way ANOVA (analysis of variance) was used with Bonferroni post-test. *p* < 0.05 was considered significant.

## 4. Conclusions

We conclude that diet should be considered as an additional important parameter for the design of experimental studies, alongside the choice of sex and strain, housing conditions, pain management and animal welfare.

## Figures and Tables

**Figure 1 ijms-22-00774-f001:**
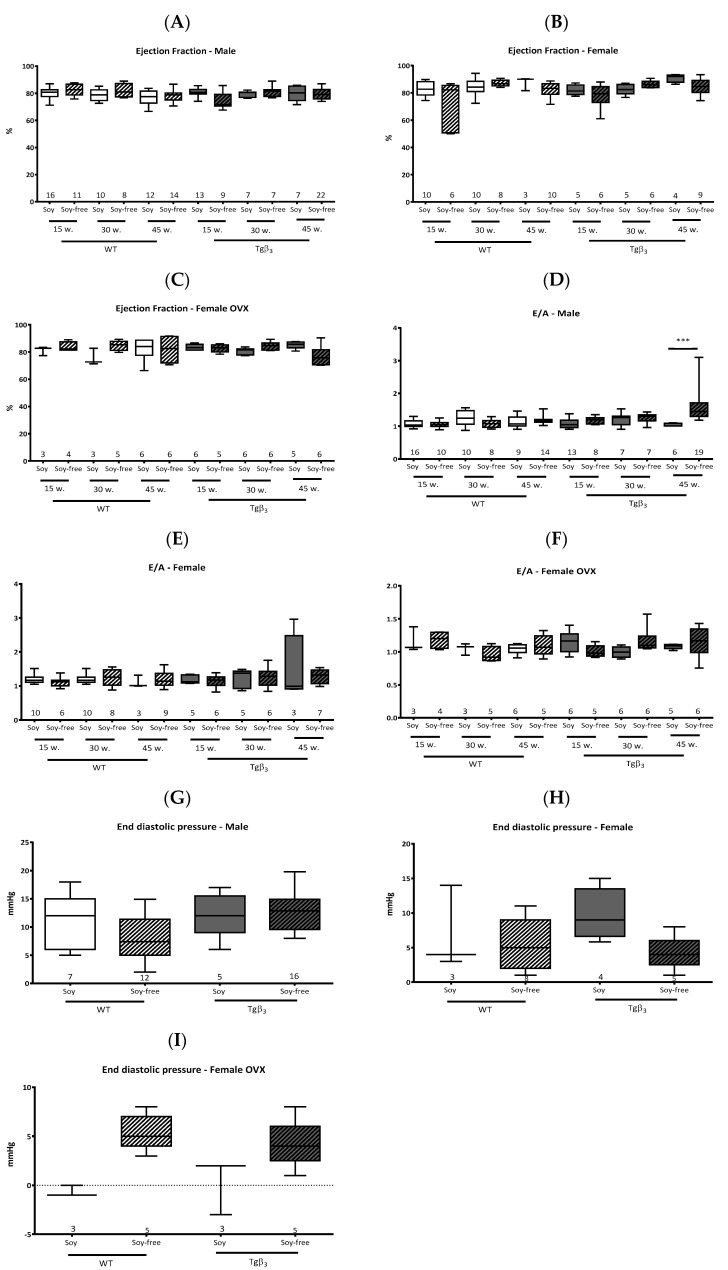
Echocardiography study at 15, 30 and 45 weeks of age and invasive hemodynamic study at 45 weeks of age depending on the rodent diet; with or without soy. We evaluated the ejection fraction on male (**A**), female (**B**) and ovariectomized (OVX) female (**C**) rats and the E/A ratio, on male (**D**), female (**E**) and OVX female (**F**) rats. Left ventricle (LV) and end-diastolic pressure (LVEDP) was evaluated at 45 weeks on male (**G**), female (**H**) and OVX female (**I**) rats. Data are expressed as mean ± SEM ***: *p* < 0.001. Echocardiography study at 15, 30 and 45 weeks of age and invasive hemodynamic study at 45 weeks of age depending on the rodent diet; with or without soy. We evaluated the ejection fraction on male (**A**), female (**B**) and OVX female (**C**) rats and the E/A ratio, on male (**D**), female (**E**) and OVX female (**F**) rats. LVEDP was evaluated at 45 weeks on male (**G**), female (**H**) and OVX female (**I**) rats. Male—WT—15 w—soy, *n* = 16; Male—WT—15 w—soyfree, *n* = 11; Male—WT—30 w—soy, *n* = 10; Male—WT—30 w—soyfree, *n* = 8; Male—WT—45 w—soy, *n* = 12; Male—WT—45 w—soyfree, *n* = 14; Male—Tgβ_3_—15 w—soy, *n* = 13; Male—Tgβ_3_—15 w—soyfree, *n* = 9, Male—Tgβ_3_—30 w—soy, *n* = 7; Male—Tgβ_3_—30 w—soyfree, *n* = 7, Male—Tgβ_3_—45 w—soy, *n* = 7; Male—Tgβ_3_—45 w—soyfree, *n* = 22; Female—WT—15 w—soy, *n* = 10; Female—WT—15 w—soyfree, *n* = 6; Female—WT—30 w—soy, *n* = 10; Female—WT—30 w—soyfree, *n* = 8; Female—WT—45 w—soy, *n* = 3; Female—WT—45 w—soyfree, *n* = 9; Female—Tgβ_3_—15 w—soy, *n* = 5; Female—Tgβ_3_—15 w—soyfree, *n* = 6; Female—Tgβ_3_—30 w—soy, *n* = 5; Female—Tgβ_3_—30 w—soyfree, *n* = 6; Female—Tgβ_3_—45 w—soy, *n* = 3; Female—Tgβ_3_—45 w—soyfree, *n* = 7; Female OVX—WT—15 w—soy, *n* = 3; Female OVX—WT—15 w—soyfree, *n* = 4; Female OVX—WT—30 w—soy, *n* = 3; Female OVX—WT—30 w—soyfree, *n* = 5; Female OVX—WT—45 w—soy, *n* = 6; Female OVX—WT—45 w—soyfree, *n* = 5; Female OVX—Tgβ_3_—15 w—soy, *n* = 6; Female OVX—Tgβ_3_—15 w—soyfree, *n* = 5; Female OVX—Tgβ_3_—30 w—soy, *n* = 6; Female OVX—Tgβ_3_—30 w—soyfree, *n* = 6; Female OVX—Tgβ_3_—45 w—soy, *n* = 5; Female OVX—Tgβ_3_—45 w—soyfree, *n* = 6. Data are expressed as mean ± SEM.

**Figure 2 ijms-22-00774-f002:**
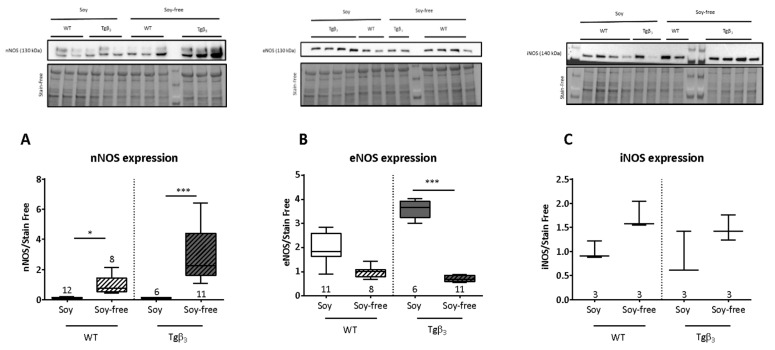
Changes in ^•^NO synthases (NOS) protein expression was performed by Western blot analysis for neuronal NOS (nNOS) protein expression. Male—WT—Soy, *n* = 12; Male—WT—Soyfree, *n* = 8; Male—Tgβ_3_—Soy, *n* = 6; Male—Tgβ_3_—Soyfree, *n* = 11 (**A**), eNOS protein expression. Male—WT—Soy, *n* = 11; Male—WT—Soyfree, *n* = 8; Male—Tgβ_3_—Soy, *n* = 6; Male—Tgβ_3_—Soyfree, *n* = 11 (**B**) and iNOS protein expression. Male—WT—Soy, *n* = 3; Male—WT—Soyfree, *n* = 3; Male—Tgβ_3_—Soy, *n* = 3; Male—Tgβ_3_—Soyfree, *n* = 3 (**C**) on LV from male rats depending on the rodent diet (with or without soy at 45 weeks). *n* = Data are expressed as mean ± SEM *: *p* < 0.05, ***: *p* < 0.001.

**Table 1 ijms-22-00774-t001:** Diet composition.

	RM1	2914C
Wheat/barley (g/kg)	885	630
Soybean (g/kg)	60	X
Whey (g/kg)	25	X
Soy oil (g/kg)	5	X
Mineral (g/kg)	25	20
Corn-derived (g/kg)	X	350
Ash (%)	6	4.5
Crude Fibre	4.65	4.1
Neutral Detergent Fibre	16.17	18
Total Saturated Fatty Acids (%)	0.51	0.6
Total Monounsaturated Fatty Acids (%)	0.88	0.7
Total Polyunsaturated Fatty Acids (%)	0.88	2.1
Metabolism energy (kJ/g)	10	12
AFE (kJ/g)	13.75	14.47
AFE from Oil	7.42	8.77
AFE from Protein	17.49	16.52
AFE from Carbohydrates	75.09	76.26
Calcium (%)	0.73	0.7
Total Phosphorus (%)	0.52	0.6
Sodium (%)	0.25	0.1
Chloride (%)	0.38	0.3
Potassium (%)	0.67	0.6
Magnesium (%)	0.23	0.2
TGE (mg/kg)	100–200	<20
Vitamin A (IU/g)	8.5	6
Vitamin D_3_ (IU/g)	0.6	0.6
Vitamin E (IU/g)	84.1	120
Vitamin K_3_ (mg/kg)	10.17	20
Vitamin B_1_ (mg/kg)	8.58	12
Vitamin B_2_ (mg/kg)	4.33	6
Niacin (mg/kg)	61.32	54
Vitamin B_6_ (mg/kg)	4.81	10
Pantothenic Acid (mg/kg)	20.17	17
Vitamin B_12_ (mg/kg)	0.007	0.003
Biotin (mg/kg)	0.27	0.26
Folate (mg/kg)	0.79	2
Choline (mg/kg)	1080	1030

AFE: Atwater Fuel Energy. TGE: Total Genistein equivalent.

## Data Availability

The data that support the findings of this study are available from the corresponding author upon reasonable request.

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
