# Peer review of "Implications of a Soy-Based Diet for Animal Models"

_ijms, 2021, doi:10.3390/ijms22020774_

Round 1

Reviewer 1 Report

While this study provides some interesting information on the impact of diet, particularly the level of dietary soy, on the potential outcome of research studies, the manuscript lacks some important details. These details include

  1. the number of rats studied in each group
  2. a more detailed description of the two diets including the type of soy or isoflavones, the type of carbohydrate (specialized often contain simple carbohydrates while standard diets often contain complex carbohydrates), levels of vitamins and micronutrients
  3. when the diets were initiated

It would also be helpful if the TgB3 rats were described in more detail. There were a few minor alterations like line 36 in the abstract where "transgenic male fed" should either be "males" or "transgenic male mice fed" and line 50 where the term "disruptor" should be replace with something more appropriate like "modifier" as dietary soy does not always disrupt endocrine signaling.

My only final concern is whether the information presented in the limited data is of significant interest to the readers of Int J Mol Sci. There are lots of studies implicating soy or isoflavones in mediating physiologic effects including heart function so it is unclear what novel information this brief study provides. 

Author Response

While this study provides some interesting information on the impact of diet, particularly the level of dietary soy, on the potential outcome of research studies, the manuscript lacks some important details. These details include:

Point 1: the number of rats studied in each group

Response 1: This information has been this information in the material and method section.

Point 2: a more detailed description of the two diets including the type of soy or isoflavones, the type of carbohydrate (specialized often contain simple carbohydrates while standard diets often contain complex carbohydrates), levels of vitamins and micronutrients

Response 2: We have added information on the composition of the food (mainly vitamins). We cannot provide more precise information on carbohydrates because the 2914C diet is patented.

Point 3: when the diets were initiated

Response 3: We thanks the reviewer for this comment. Diet start in each group when the rat are 8 weeks of age, this information has been added this information in the material and method.

Point 4: It would also be helpful if the Tgβ3 rats were described in more detail.

Response 4: The Tgβ3 model has been described in a recently published article, this article details the phenotype observed in rats [1]. We add some information in the material and method.

Point 5: There were a few minor alterations like line 36 in the abstract where "transgenic male fed" should either be "males" or "transgenic male mice fed" and line 50 where the term "disruptor" should be replace with something more appropriate like "modifier" as dietary soy does not always disrupt endocrine signaling.

Response 5: We thank the reviewer for these suggestions. We have made the changes in the article.

Point 6: My only final concern is whether the information presented in the limited data is of significant interest to the readers of Int J Mol Sci. There are lots of studies implicating soy or isoflavones in mediating physiologic effects including heart function so it is unclear what novel information this brief study provides.

Responses 6: We believe that this observation is of great importance for any reader trying to develop animal model and experiencing month of negative results.

Reviewer 2 Report

In this article, Dhot et al investigated the effect of the diet and more especially the effect of diet deprived of soy on the appearance of diastolic dysfunction in a rat model of diastolic dysfunction induced by overexpression of the b3 receptor in the endothelium. This is a very important question rarely taken into account and which joins the importance of sex in the development of pathologies.

The paper is well written and organized however the figures are really difficult to read.

Main suggestions:

1) As your goal is to compare the two different diets, you should present your figures differently. You should compare soy and soy-free for wild type, soy and soy free for transgenic for the different ages and groups. This should be done for the two figures. In addition, please enlarge the lettering of the figures.

2) As you performed your statistical analysis by two-way ANOVA comparing genotype and diet please provide the results for genotype effect, diet effect and possible interactions between both.

3) The discussion should be developed according to the actual knowledge in the field of estrogens and estrogen receptors effects on cardiacvacular pathophysiology. As you state in your discussion soy is a phytoestrogen that mimics natural estrogens. You should thus discuss your results in light of the known effects of estrogens on NOS expression or on end-diastolic function etc…

Round 2

Reviewer 1 Report

Point 1. listing the number of animals as n=3-22 is not very helpful. The number of mice used in each experiment should be explicitly stated in the results or figure legends

Point 2. I have worked with a number of proprietary diets and I would be surprised if the company that supplied the diet would not give you the source of carbohydrates. I have always been able to get this information.

Point 6. The authors did not detail how their study advanced the field.

Round 3

Reviewer 1 Report

The comments have been addressed